# Antioxidant Activity of Mushroom Extracts/Polysaccharides—Their Antiviral Properties and Plausible AntiCOVID-19 Properties

**DOI:** 10.3390/antiox10121899

**Published:** 2021-11-26

**Authors:** Sechul Chun, Judy Gopal, Manikandan Muthu

**Affiliations:** Department of Environmental Health Science, Konkuk University, Seoul 143-701, Korea; scchun@konkuk.ac.kr (S.C.); jejudy777@gmail.com (J.G.)

**Keywords:** mushrooms, polysaccharides, antioxidant, antiviral, antiCOVID-19, bioactivity

## Abstract

Mushrooms have been long accomplished for their medicinal properties and bioactivity. The ancients benefitted from it, even before they knew that there was more to mushrooms than just the culinary aspect. This review addresses the benefits of mushrooms and specifically dwells on the positive attributes of mushroom polysaccharides. Compared to mushroom research, mushroom polysaccharide-based reports were observed to be significantly less frequent. This review highlights the antioxidant properties and mechanisms as well as consolidates the various antioxidant applications of mushroom polysaccharides. The biological activities of mushroom polysaccharides are also briefly discussed. The antiviral properties of mushrooms and their polysaccharides have been reviewed and presented. The lacunae in implementation of the antiviral benefits into antiCOVID-19 pursuits has been highlighted. The need for expansion and extrapolation of the knowns of mushrooms to extend into the unknown is emphasized.

## 1. Introduction

Mushrooms belong to the Basidiomycetes group of macrofungi. Mushrooms can grow either above the soil (epigeous), or below the soil (hypogeous). Mushrooms are the choice ingredients of gourmet cuisine globally, encompassing a unique flavor, that works culinary wonders. A total of 2000 species of mushrooms exist, 25 of which are accepted as food and few are commercialized. Mushrooms are also known for their nutritional, organoleptic merits and medicinal properties [1,2]. Their therapeutic qualities, although known much earlier, have recently been acknowledged and valued. The healing properties of mushrooms were known in Chinese traditional medicine even as much as thousands of years before and they are still being used today [3]. Mushrooms abound in essential amino acids, minerals, proteins, and biologically active polysaccharides. They are predominantly consumed in Asian countries, however, in recent years, *Pleurotus ostreatus*, *Boletus edulis*, *Lentinula edodes* (Shiitake), *Ganoderma lucidum* (Reishi), *Trametes versicolor*, *Grifola fronda* (Maitake), *Agaricus bisporus* and *Agaricus subrufescens*, have been widely popularized worldwide [4,5].

Mushrooms have the inherent ability to accumulate minerals and vitamins and various other secondary metabolites, such as organic acids, alkaloids, phenols and terpenoids [6]. The production of mushrooms has been continuously on the increase, China being the largest global producer [1,7,8]. Wild mushrooms also have their own popularity and nutritional, sensory and pharmacological attributes [2]. Mushrooms are an alternative source of new antimicrobial compounds, terpenes, steroids, anthraquinones, quinolones and benzoic acid derivatives, as well as oxalic acid, peptides, and proteins (primary metabolites. Edible mushrooms possess B1, B2, B12, C, D, and E tannins of nutritional significance [9,10], becoming a rich reservoir of diverse nutraceuticals displaying the synergistic effects of multiple bioactive compounds [11,12,13,14,15].

The pharmaceutical potential of mushrooms has in the last few decades escalated, and as of now mushrooms are realized and branded as mini-pharmaceutical factories [8,16]. The contents of biologically active substances may vary considerably, affected by variations in strain, substrate, cultivation, developmental stage, age, storage conditions, processing, and cooking practices [12,13,17]. However, whatever the case, there is no dearth for bioactive substances in mushrooms. The exhaustive list of acids, terpenoids, sesquiterpenes, polyphenols, lectins, alkaloids, lactones, sterols, metal chelating agents, nucleotide analogs, vitamins, glycoproteins, ergosterols, volatile organic compounds and polysaccharides are there as always.

The current review will focus on the briefly reviewing the biological activity of mushrooms followed by reviewing the antioxidant activity of mushrooms. The antiviral properties of mushrooms, specifically the antiCOVID-19 activity of mushrooms has been elaborately dealt with. The need for improvising the gaps in the proper utilization of this natural resource for positive outcomes has been discussed as a future outlook.

## 2. A Snapshot of the Biological Activities of Mushrooms

Mushrooms are responsible for over a hundred odd medicinal functions. Its key medicinal applications include: antioxidant, anticholesterolemic, anticancer, antidiabetic, antiallergic, immune modulating, cardiovascular protection, antiviral, antibacterial, antiparasitic, anti-inflammatory, antifungal, detox and hepatoprotective effects [18,19,20,21,22,23]. Various mushroom extracts could mediate decreased activity of inflammatory mediators (nitric oxide (NO), cytokines, and prostaglandins), reducing cell inflammations. Various mushroom extracts have been successfully demonstrated for: immune modulator [24,25,26,27], antitumor/anticancer [28,29,30,31,32], antibacterial and antiviral [33,34,35], antioxidant [36,37,38,39], and antihypoglycaemic [40,41,42] applications and as antiatherosclerotic agents [43]. Shaffique et al. have very recently reviewed the antioxidant attributes of medicinal mushrooms [44]. The efficacy of the bioactive compounds can be largely influenced by the mushroom type, substrate, cultivation conditions and fruiting conditions, stage of development, age, storage conditions and processing and cooking methods [43].

The anticancer milestones of mushrooms have been elaborately reviewed by Ren et al. [45]. We will highlight a few major outlines. *Agaricus silvaticus* mushrooms, when supplemented in food, reduced glycaemia levels in cancer patients [46] and proved beneficial in patients with colorectal cancer (postsurgery) [43]. A polysaccharide from *Grifola frondosa* hyped the immune system, when administered orally to breast cancer patients [47]. Japanese researchers confirmed that oral administeration of *Lentinula edodes* mycelial extracts helped Japanese chemotherapy patients [48,49,50,51,52] and β-glucan lentinan increased the lifetime of advanced gastric cancer patients [53]. *Agaricus silvaticus* reduced nausea and abnormal bowel symptoms in those subjected to chemotherapy for breastcancer [54]. A meta-analysis suggested that mushroom intake reduced the incidence of breast cancer [55]. Consumption of mushrooms prevented colitis-associated cancer by reducing cell proliferation and mucosal inflammation [56]. Oral intake of powdered *Agaricus bisporus* by prostate cancer patients influenced prostate-specific antigens (PSA) and altered the etiology of recurrent prostate cancer through its immuno modulating activity [57,58,59].

Holmes [60] and Chang et al. [61] confirmed the antiobesity activity of *Ganoderma lucidum* in mice by altering gut microbiota composition. In a clinical trial in 2009, Johnson et al. [62] confirmed that a daily oral dose of AndoSan (a mushroom extract mixture containing *A. blazei* mycelium 82%, *Hericium erinaceum* 15%, *Grifola frondosa* 3%) when administered to subjects for 12 days, led to significant in vivo reduction in interleukin-1 (IL-1). The genus Cordyceps includes *C. sinensis* and *C. militaris* which are the most valued species in Traditional Chinese Medicine [63]. These have been established for immunological regulation, free radical scavenging, anticancer, antimicrobial, analgesic, antihyperlipidemic, antileukemic and lung improving attributes. The immunomodulatory active substances from mushrooms stimulate immune effector T cells. Cytotoxic dendritic cells (DCs), lymphocytes, T lymphocytes (CTL), macrophages and natural killer (NK) cells, resulting in cytokine expression and interleukins (ILs), tumor necrosis factor-alpha (TNF)-α, and interferon-gamma (INF)-γ are stimulated by bioactive substances from various mushrooms [64,65]. Other immunomodulators like lectins, proteins, polysaccharides and terpenoids are also known [66]. Triterpenoids such as lanostane present in wood-decaying mushrooms, Ganoderma sp, exhibiting immunomodulating and anti-infective effects [67,68,69]. *G. lucidum*, *Grifola frondosa*, *Flammulina velutipes*, *Agaricus bisporus*, *Agaricus blazei, Coprinus cinereus*, *Cordyceps sinensis*, *Laetiporus sulphureus*, *Lentinus tigrinus*, *Trametes versicolor*, *Amanita pantherina*, *Boletus satanas*, *Ischnoderma resinosum*, *Lactarius deterrimus* and *Volvariella volvacea* are reputed for their immunomodulatory activities [70,71,72,73,74,75]. Mushrooms can act as adaptogens and immunostimulators, and their immunostimulatory property is primarily prophylactic and non-invasively prevents infectious diseases and tumor metastases.

The genus Pleurotus has several species that produce mevinolin [76]. Oyster mushroom produced lovastatin; when 5% of the dried oyster mushroom fruiting bodies was added to a high-cholesterol diet, cholesterol was significantly reduced. Mushrooms are able to redistribute cholesterol in favor of high-density lipids (HDL), reduced production of total cholesterol (TC), very-low-density lipoprotein (VLDL), low-density lipoprotein (LDL), reduced cholesterol absorption and β-hydroxy β-methylglutaryl-CoA (HMG-CoA) reductase activity in the liver [77]. Mushrooms are also well known for their antimicrobial activity, *L edodes* exhibits antimicrobial action against both Gram-positive and Gram-negative bacteria [78] and various other mushroom species have been well established in terms of this attribute.

## 3. Antioxidant Activity of Mushrooms

Researchers have established the fact that the antioxidant activity of mushroom is the genesis of a plethora of bioactivities. Antioxidant compounds have been extracted from fruiting bodies, mycelium and broth of various mushrooms [79]. Mushroom components that are reputed for their strong antioxidant properties include: phenolics, flavonoids, glycosides, polysaccharides, tocopherols, ergothioneine, carotenoids, and ascorbic acid [80,81,82,83,84,85,86,87,88,89,90,91,92,93,94,95,96,97,98,99,100,101,102,103,104,105,106,107,108,109,110,111,112,113,114,115,116,117,118,119,120,121,122,123,124,125,126,127,128,129,130,131,132,133,134,135,136,137,138,139,140,141,142,143,144,145,146,147,148,149,150,151,152,153,154,155,156,157,158]. These antioxidant compounds from mushrooms have been identified and quantified using high performance liquid chromatography (HPLC) and gas chromatography (GC), nuclear magnetic resonance (NMR), Fourier transform infrared (FT-IR), UV-VIS spectroscopy and various spectrophotometric assays [12,82,83,84,85,86,87,88,89,90,91,92,93,94,95,96,97,98,99,100,101,102,103,104,105,106,107,108,109,110,111,112,113,114,115,116,117,118,119,120,121,122,123,124,125,126,127,128,129,130,131,132,133,134,135,136,137,138,139,140,141,142,143,144,145,146,147,148,149,150,151,152,153,154,155,156,157]. The antioxidant potential of mushrooms is a well-accepted fact. The antioxidant activity of mushroom extracts is measured using methods based on the transfer of electrons and hydrogen atoms, the ability to chelate ferrous (Fe^2+^) and cupric (Cu^2+^) ions, the electron spin resonance (ESR) method, erythrocyte hemolysis, and the monitoring of the activity of superoxide dismutase (SOD), catalase (CAT) and glutathione peroxidases (GPx) [155,156,157]. Rufoolivacin, rufoolivacin C, rufoolivacin D and leucorufoolivacin have been demonstrated for their ability to scavenge DPPH radicals [157,158]. *Ramaria flava* phenolics aid in scavenging of 2,2-diphenyl-1-picrylhydrazyl (DPPH) and hydroxyl (OH) radicals [159,160]. Ferreira et al. have published an excellent review on the antioxidants in wild mushrooms [13]. This review chooses to highlight the biological impacts of mushroom polysaccharides and their antioxidant activity.

### 3.1. Bioactivity of Mushroom Extracts and Their Polysaccharides

*Pleurotus spp.* have a vast diversity of polysaccharides, particularly heteropolysaccharides and glucans [161]. These polysaccharides from the genus *Pleurotus spp.* are known to carry special biological activities. Ruthes et al. (2016), in their review article, have reported that mushrooms are abundant in heteropolysaccharides [4]. Heteropolysaccharides possess anti-tumor, antioxidant, anti-inflammatory, and immunomodulatory activity. Barbosa et al. (2020b) employed a special supercritical binary hot water and CO_2_ system to extract polysaccharide-rich fractions from *Pleurotus ostreatus* [162] and their antioxidant potential has been demonstrated in cell models. *Phallus atrovovatus* is known to possess abundant polysaccharides, predominantly fractions of β-glucan and α-glucan. These polysaccharides exhibited immune system modulating activity and high anti-inflammatory activity [163].

Mingyi et al. (2019) showed that mushroom polysaccharides are high functionality biomolecules [164]. The review consolidates the use of polysaccharides in foods, as medicines and in cosmetics and its future as a functional food. Polysaccharides have now fully recognized as the major bioactive components of mushrooms [165,166], which are bound to the mushroom cell wall by covalent (ester) linkages [84]. As already specified, the most widely reported activity of mushroom polysaccharides is antioxidative [82,83,85,87,88,89,90,91,92,93,94,95,96,97,98,99,100,101,102,103,104,105,106,107,108,109,110,111,112,113]. Briefly, we now review and present a consolidated account of the various reports on the antioxidant activity of mushroom polysaccharides published thus far. It is believed that purified mushroom polysaccharides exhibit lower antioxidant activities than their original crude extracts [95], while others reported high antioxidant activity in pure polysaccharide fractions. *A. brasiliensis* polysaccharides (consisting mainly of (1→6)-β-d-glucans) obtained by pronase deproteinization exhibited high antioxidative activity against ^•^OH and O_2_^•−^ radicals [109]. β-glycans are the predominant antioxidative components in mushrooms and are responsible for activating systemic responses [167,168]. Thus, they are the jackpots of mushroom polysacchrides in terms of their bioactive potential. Mushroom polysaccharides and glycoconjugates are now becoming ideal candidates for creating new nature-based medications, for dietary supplements and for treatment of oxidative stress-mediated disorders. Polysaccharides also help prevent lipid peroxidation and the pathogenesis of various gastro intestinal (GI) diseases, such as peptic ulcers, GI cancers and inflammatory bowel disease which stem from oxidative stress [169,170].

Mushroom polysaccharides also display antimicrobial properties against pathogenic bacteria and viruses. Data on mushroom polysaccharides for different basidiomycetes indicated the presence of rhamnose, xylose, fucose, arabinose, fructose, glucose, mannose, mannitol, sucrose, maltose and trehalose as the predominant mushroom-based polysaccharide fractions [171]. Klaus et al. [86] encapsulated polysaccharide extracts from *G. frondosa* in alginate gel beads to protect them from external influences and extend their applications. Mushroom polysaccharides have been extracted from: *A. blazei* [172], *A. brasiliensis* [173], *A. ponderosa* [174], *oyster mushroom* [175], *A. polytricha* [176], *B. edulis* [177], *C. tricholoma* [178], *C. militaris* [179], *Entolomalivido album* [180], *Gleoestereum incarnatum* [181], *G. lucidum* [182], *Grifola frondosa* [183,184], *Hohenbuehelia serotina* [185], *Hypsizygus marmoreus* [186], *Iliodiction cibarium* [41,45], *Lactarius deliciosus* [187], *L. edodes* [188], *Macrolepiota dolichaula* [189], *Phellinus igniarius* [190], *Phellinus linteus* [191], *Phellinu spini* [192], *Pholiota adiposa* [193], *Pholiota nameko* [194], *Pleurotus eryngii* [195], *P. ostreatus* [196], *Termitomyces heimii* [197], *Tricholoma matsutake* [198,199,200,201], *Tricholoma mongolicum* [202].

Likewise, mushroom polysaccharides are becoming increasingly well suited against obesity. Combination cancer therapy using a *Grifola frondosa* β-glucan fraction and an oligodeoxynucleotide is reported [203]. Pan et al. [204] showed that *Amauroderma rude* polysacchrides inhibit tumor in mice. Encapsulation of *Antrodia camphorata* polysaccharides in chitosan–silica/silica nanoparticles increased the anti-tumor activity of HepG2 liver cancer cells [205]. Polysaccharide-contents of *Hericium erinaceus* extracts inhibited migration of cancer from colon tumors to lungs in murine models [206]. *Lentinula edodes* enhanced immunity in healthy young people and oral intake of soluble β-glucans in elderly healthy adults increases the number of circulating β-cells [207]. Meng et al. [208] and Yan et al. [209] have elaborately discussed the link between structural characteristics of mushroom polysaccharides and their other relational antitumor aspects. Schwartz and Hadar [56] have reviewed the possible mode of action of mushroom β-glucans against cancer associated with inflammatory bowel disease in humans. Mushroom polysaccharides seem to orchestrate antitumor properties via activation of the host immune response. Thus, it is inferred that mushroom polysaccharides do not directly destroy tumor cells; instead, they indirectly make an impact by preventing stress on the body, leading to a 50% reduction in the size of the tumor, thereby prolonging the survival time of the tumor-bearing mice [51,177].

Supplementation with β-glucan from *Pleurotus ostreatus* is reported to protect athletes from respiratory tract infections [210]. Silver nanoparticles were synthesized using glucan from *Pleurotus florida*. This could inhibit *Klebsiella pneumoniae* synergistically (along with four medicinal antibiotics) [211]. *Pleurotus nebrodensis* polysaccharide enhanced immunity and inflammatory responses by activating macrophages [37]. An exopolysaccharide from *Clitocybe maxima* increased the immunological response and inhibited tumor cells in mice [64,212]. Manna et al. [198] synthesized nanoparticles using a *Lentinus squarrosulus* hetero polysaccharide and successfully demonstrated it against *E. coli* and other bacteria. The nanoparticles were better than normal particles in inhibiting bacteria and viruses. Mushroom polysaccharides shielded mice against Salmonella lipopolysaccharide-induced septic shock [213]. The same polysaccharide combined with *Hericium erinaceus* extracts protected mice against *Salmonella typhimurium* by stimulating the immune system [214]. *Lentinus edodes* extracts were demonstrated against oral pathogens [215], while *Auricularia auricula-judae* crude polysaccharides were active against *Escherichia coli* and *Staphylococcus aureus* [216]. A sulphated polysaccharide from oyster mushrooms showed antibacterial activity against foodborne *E. coli* and *Staphylococcus aureus* [217]. *Polysaccharides* from *Hericium erinaceus* have been of considerable interest due to their antioxidant activities [218]. Polysaccharides from eight *Hericium* species exhibited significantly high antioxidant activity and inhibited proliferation of tumor cells [219]. In another report, bismuth–polysaccharide complexes could inhibit *Helicobcter pylori*, which cause human ulcers and eventual cancers [220].

Huang et al. [221,222] showed that *Pleurotus tuber-regium* polysaccharides exhibited antihyperglycemic properties and lowered oxidative stress in diabetic rats. *Agaricus blazei* polysaccharide extracts impacted proinflammatory cytokine production in human monocytes and endothelial cells [223] and other bioactive fractions as well (IL-6, prostaglandin D(2), leukotriene C(4)). *L. edodes* active fucomannogalactan fractions of (1→6)-linked main chains have shown anti-inflammatory activities in male Swiss mice [202]. The polar fraction the β-glucan-rich mushroom preparation AndoSan™ showed antitumor activity in RAW 264.7 macrophage cells. β-glucans show anti-asthma and antitumor properties, as well as anti-inflammatory effects in inflammatory bowel disease [224]. With all these added assets, nevertheless a study where AndoSan™ was administered to 40 patients with multiple myeloma due for chemotherapy, no significant responses were observed [225]. This indicates that there is a long way to go for all these to be worked out in order to practically demonstrate statistically significant responses in real time.

*Agaricus blazei* extract increased immune response against foot-and-mouth disease [226] and *H. erinaceus* extracts shielded the mice against *Salmonella typhimurium* [227] and *L. edodes* mycelial polysaccharids defended mice against Salmonella-induced endotoxemia and salmonellosis [216]. Another study by Kim et al. confirmed the protective effect of *C. sinensis* extracts on the lipopolysaccharide induced lung injury in mice. The extracts reduced TNF-α, IL-6, IL-1β expression, as well as the binding ability of NF-κB p65 DNA and inhibitied the mRNA expression of cyclooxygenase (COX)-2 and inducible nitric oxide synthase (iNOS) in lung tissue [228]. *C. sinensis*, was proposed by the same authors as for treatment of acute lung injury. Yang et al. [229] reported the use of *C. sinensis* towards the inhibition of bleomycin-induced fibrosis in mice, to prevent and treat lung fibrosis. Mueller et al. [230] and Jiang et al. [231] have confirmed that *C. sinensis* inhibits lung fibrosis. These results are supported by Lee et al. [232], who showed that cordycepin from *C. militaris* downregulates iNOS, COX-2 expression and TNF-α gene expression. Additionally, Ohta et al. [233] showed that cordycepin reduced Th2 associated cytokines, including IL-4, IL-5 and IL-13, in Ova-induced asthma mice.

A total of 14 species of Basidiomycetes mushroom cultures were studied. All these species showed 20% more antioxidant potential. [234]. Other authors quantified the antioxidant potential of *Ganoderma lucidum*, *Ganoderma tsugae*, *Coriolus versicolor*, *G. tsugae* and *G. lucidum* methanolic extracts. Their antioxidant activity is driven by their phenolic contents. [235]. In 2008, Kim et al. investigated the antioxidant potential of edible medicinal mushrooms, *Agaricus bisporus* showed the highest activity. They reported a 78% positive correlation between phenolics and antioxidant potentials [236]. The antioxidant potentials of methanolic extracts of shiitake and oyster mushrooms using the 1,3-diethyl-2-thiobarbituric acid method were reported to be due to their phenolic contents [237]. The antioxidant potentials of five methanolic extracts of ear mushrooms, including red, black, jin, snow, and silver ear mushrooms were determined. The methanolic extracts contained bioactive tocopherol, polyphenols and ascorbic acid and the snow ear variety possessed maximum antioxidant potential [105].

The antioxidant potential of ethanolic extracts of *Laetiporus sulphureus* was studied. A positive correlation between polyphenol contents and antioxidant potential was observed. The antioxidant potential of various *Morchella sps and Meripilus giganteus*, *Armillaria mellea*, *Paxillus involutus*, *Pleurotus eryngii*, and *Pleurotus ostreatus*, via the DPPH method was measured. Among these, *M. elata* possessed the maximum antioxidant potential toward free radicals [238]. Methanolic extracts of *Inonotus obliquus* also possessed strong antioxidant potentials owing to their polyphenols, such as inonoblins A–C and phelligridins D, E, and G. [239]. The antioxidant and hepatoprotective pattern of *Lentinus edodes* was validated in an independent study, using mice models [240].

An in vitro study conducted in Iran confirmed the antioxidant potential of *Cantharellus cibarius* and *Pleurotus porrigens* methanolic and ethyl acetate extracts, via the DPPH method [241]. Hot water extracts of Agaricus, Antrodia, Auricularia, Coprinus, Cordyceps, Hericium, Grifola, Ganoderma, Lentinus, Phellinus, and Trametes were tested for their antioxidant potential. Polyphenolic compounds and polysaccharides were responsible for the high antioxidant potential of all these studied mushrooms. Among all, Ganoderma was the most antioxidant mushroom [103]. The antioxidant activities of two cultivated mushrooms—*P. ostreatus* and *L. edodes*—and five other wild mushrooms from Ethiopia were compared. Results indicated that *A. campestris* exhibited significant antioxidant potential due to its phenolic compounds [103]. The Polyporoid species of medicinal mushrooms native to Poland were studied. The results showed that it contains protocatechuic, vanillic, and hydroxybenzoic acids and that phenolic compounds were behind their antioxidant potentials.

The antioxidant potential of *Pleurotus eryngii*, due to its phenolic contents, revealed that it has excellent antioxidant activity and was able to scavenge free radicals and possessed reducing power. It also contained ergothioneine, making it a successful functional food [242]. *G. lucidum* was observed to possess high phenolic contents and significant antioxidant activity and potential as a good functional food [243]. *Ganoderma tsugae* showed high antioxidant levels owing to its phenols [244]. For the Leucopaxillus species, negative linear regressions were seen between flavonoids, which increased with the antioxidant activity [245]. *Pleurotus ferulae*, *Clitocybe maxima*, and *Pleurotus ostreatus* were selected for antioxidant study and their antioxidant potentials measured. The results showed that they contained phenolic compounds that helped them fight the oxidative stress system [246]. Methanolic extracts of *Pleurotus porrigens* and *Hygrocybe conica* indicated that *Hygrocybe conica* possessed higher chelating and antioxidant properties due to their total phenolic components [247]. Other authors have reported the antioxidant and the anti-inflammatory effects of Malaysian *G. lucidum* aqueous extracts; their study proved that these extracts exhibited higher antioxidant and anti-radical effects [248,249]. We will now elaborate on the reports on how mushroom polysacchrides antioxidants have been used in various applications.

#### 3.1.1. Antioxidant Mushroom Extracts and Polysaccharide Applications

Most mushrooms contain polysaccharides that include chitin, glucans and heteroglycans. These polysaccharides are instrumental in coordinating the growth and developmental processes of the mushroom’s fructiferous body. Polysaccharides play an important role in modulating the immunity of human cells [250]. Immunomodulating polysaccharides such as β-glucans are non toxic and have no secondary effects while being used against bacteria and viruses [45]. They also display antitumor and immunostimulating properties [251]. The antioxidative and immunostimulating properties of *Cordiceps militaris* polysaccharides were able to suppress the in vivo growth of melanoma in mouse models [252]. Antioxidative properties of a crude polysaccharide from *Inonotus oblique* (folk medicine in Russia)include rich medicinal and nutritional properties [253]. A polysaccharide from *Hericium erinaceus* exhibited strong in vitro antioxidant activity and liver damage protection [254]. *Macrolepiota dolichaula* fucogalactan [193] and β-glucan from *Russula albonigra* [255] showed excellent antioxidant activity. *H. erinaceus* polysaccharide exhibited antioxidant and neuroprotective effects [256]. A polysaccharide from *Agaricus brasiliensis* induced immunostimulation and cell proliferation in vitro in mice [257,258]. Polysaccharides extracted from *Tricholoma mongolicum* displayed in vitro antioxidant activity [201]. Ultrasonically extracted Ganoderma β-d-glucans were reported to possess better in vitro antioxidant activity than conventionally extracted ones [191,259], owing to the fact that the ultrasonic extraction preserved their molecular weights and degree of branching. A water-soluble β-glucan, isolated from the fruit bodies of *Entoloma lividoalbum*, stimulated the production of macrophages, splenocytes and thymocytes and exhibited hydroxyl and superoxide radical scavenging activities and reducing properties [260]. A fucogalactomannan from *Tylopilus ballouii* mushroom inhibited superoxide and hydroxyl radicals and reduced edema [261].

In another study, polysaccharides from eight Chinese mushrooms were evaluated for their total carbohydrate, polyphenolic and protein contents, and antioxidant and anti-proliferation activities. The results suggested that all the polysaccharides had significant antioxidant capacities. The acid extracts of *Russula vinosa* had the highest ABTS^+^ scavenging activity, and *Dictyophora indusiata* and *Hohenbuehelia serotina* possessed the highest ^•^OH scavenging capacity and ability to inhibit lipid peroxidation [262].

M. Kozarski compared polysaccharides of *A. bisporus*, *A. brasiliensis*, *Phellinus linteus*, and *G. lucidum* and their other bioactive components. A positive correlation between glucan level and antioxidant activity was reported in case of *G. lucidum*. [84]. A study conducted by The University of Calcutta revealed that *Pleorotus squarrosulus*, *Fistulina hepatica*, *Austreus hygrometricus*, *Polyporus grammocephalus*, *Phellinus linteus* and *Macrocybe gigantea*, have high antioxidant activity [263]. Methanolic extracts of the wild *Ganoderma lucidum*, native to the Himalayas, showed significant antioxidant potential [264]. The antioxidant potentials of *Volvariella volvacea* [265], *Ganoderma tsugae*, *Morchella conica*, [266], *Ganoderma lucidum*, *Hypsizygus marmoreus*, *P. ostreatus*, *P. nebrodensis*, *Lentinus edodes*, *Pleurotus eryngii*, *Flammulina velutipes*, and *Hericium erinaceus* are clearly documented [249,267]. Further study was also conducted to assess the effectiveness of medicinal mushrooms on MCF-7 breast cancer cell lines. The results showed that *G. lucidum* polysaccharides wroked well against MCF-7 cell lines [221]. The antioxidant potential of Taiwanese *Cordyceps taii* is also reported [268].

#### 3.1.2. Mechanism of Antioxidant Mushroom Polysaccharides

The antioxidant properties of mushrooms are related to the bioactive compounds in mushrooms. Mushrooms are the primary source of ergothioneine, which protects the mitochondrial components from oxidative damage. This is orchestrated by generation of O_2_^•−^ through the escape of electrons from the mitochondrial electron transport system (ETS) [269]. The antioxidative activity of mushroom polysaccharides is attributed to their RS scavenging ability, reduction property and ability to chelate Fe^2+^, inhibit lipid peroxidation, erythrocyte hemolysis and the increase in enzyme activities in eukaryotic and prokaryotic cells and their roles in ongoing SOD, CAT and GPx antioxidative processes [270].

The potency of mushroom polysacchrides to scavenge free radicals is owing to the presence of hydrogen in certain monosaccharide units and their binding in side branches of the main chain [94,121]. The enhanced antioxidant activity is owing to the abstraction of the anomeric hydrogen from one of the internal monosaccharide units rather than from the reducing end [271]. Recently, Kishk and Al-Sayed [272] reported that the ^•^OH scavenging mechanism of polysaccharides is same as that of phenol compounds. The mechanism is dictated by hydrogen atom transfer (HAT) reactions. These reactions mostly occur in neutral polysaccharides, while the electron transfer (ET) mechanism usually occurs in acidic polysaccharides. Mushroom antioxidants act in varying stages and via different mechanisms [13,273]. There are two main types of mushroom antioxidants, namely, primary (chain breaking, free radical scavengers) and secondary or preventive [84,97,103,125,136,152,153,154] antioxidants. Secondary antioxidants manifest from deactivation of metals, inhibition or breakdown of lipid hydroperoxides, regeneration of primary antioxidants, or singlet oxygen (1O_2_) quenching processes [80]. In certain other cases, mushroom ROS scavengers act in oxidation–reduction reactions that are reversible, and as antioxidants and pro-oxidants. The optimization of antioxidant dietary supplements from mushrooms is far from accomplished [17,42,185]. Table 1 summarizes the list of mushroom polysaccharides and their antioxidant activity.

Figure 1 enlists the known mechanisms of antioxidant activity of mushroom polysaccharides elucidated to date.

## 4. Antiviral Activity of Mushrooms/Mushroom Polysaccharides

The antiviral activity of mushrooms is another add-on to the exhaustive bioactivities of mushrooms. Wild mushrooms from Russia, such as *Daedaleopsis confragosa*, *Datronia mollis*, *Ganoderma valesiacum*, *Irpex lacteus*, *Ischnoderma benzoinum*, *Laricifomes officinalis*, *Lenzites betulina*, *Phellinus conchatus*, *Piptoporus betulinus*, *Trametes gibbosa*, and *Trametes versicolor* have been shown to have antiviral activity on A/chicken/kurgan/05/2005 (H5N1) bird virus and the A/Aichi/2/68 (H3N2) human virus. The report confirmed that these mushrooms produced antiviral substances that block the synthesis of viral enzymes and boost human immunity [277]. Another study in Russia in 2020, by Ilyicheva et al., assessed the antiviral effect of ethanol and water extracts of *Pleurotus pulmonarius* fructiferous body against the A/California/07/09(H1N1pdm) virus, the results showing that the ethanol extracts had a more powerful antiviral effect than the water extracts. This mushroom’s fructiferous body is an important source of polysaccharides, which are responsible for the inhibitory activity against infections caused by this flu virus [278]. The water extract of *Pleurotus tuber-regium*, containing β-glucans, was demonstrated against herpes simplex virus type 1 (HSV-1), herpes simplex virus type 2 (HSV-2), respiratory syncytial virus (RSV) and influenza A virus (Flu A), which was tested by Zhang et al. in 2004 [279]. The tests confirmed powerful antiviral effect against HSV1 and HSV-2. Water and methanolic extracts of *Boletus edulis*, *Lentinus edodes* and *Pleurotus ostreatus* were tested against the herpes simplex type 1 (HSV-1) viruses and water extracts were observed to show high antiviral activity. The highest antiviral effect was found in *Lentinus edodes* extracts, followed by *Boletus edulis* and finally by *Pleurotus ostreatus* [280].

*Agaricus brasiliensis*’ antiviral activity was proven by Faccin et al., where polysaccharides from water and ethanol extracts were successfully demonstrated against type 1 poliovirus, which is one of the mushroom species that grows on tree stumps, having been domesticated in order to be cultivated [281]. In 2007, Gu et al. [282] and in 2016 Zhao et al. [283] evaluated the antiviral activity of *Grifola frondosa* against enterovirus 71 and HSV-1. These reports confirmed the antiviral effects of polysaccharides and their potent use as therapeutic antiviral agents. *Inonotus obliquus* is a parasite mushroom that lives up to 20 years; in 2011, Shibnev et al. used water extracts of this mushroom against hepatitis C virus [284]. The extracts inhibited the infectious viruses in the kidney cells of a pig embryo. Methanolic extracts from *Pleurotus sp.* and *Lentinus sp.* were used against cytomegalovirus (HCMV) [285]. In another study, β-glucans of *Pleurotus ostreatus* were used to control influenza viral infection of the respiratory tract in children [286].

Some HIV-1 protease inhibitors have been isolated from medicinal mushrooms. Various components that possess anti HIV-1 protease activity have been isolated from *Ganoderma lucidum* (ganolucidic acid A, 3 β-5α-dihydroxy-6β-methoxyergosta-7,22-diene, ganoderic acid A–C, ganoderic acid β, ganodermanondiol, ganodermanontriol and lucidumol B) [287,288,289]. Six colossolactones, ganomycin I, and ganomycin B, isolated from *G. colosum*, with anti-HIV-1 protease activity have been reported, as also in *G. sinnesse* [290,291]. Tiger milk mushroom (*Lignosus rhinocerus*) and *Auricularia polytricha* also exhibited HIV-1 protease activity [292].

Adenosine and iso-sinensetin isolated from *Cordycep militaris* and 4.5 kDa protein isolated from *Russula paludosa* have been demonstrated to have anti-HIV-1 protease activity [293]. *C. sinensis* and *C.militaris* are known to exhibit antiviral effect on several viruses. In 1991, Mueller et al. [230] reported the in vitro antiviral effect of cordycepin on HIV-1. Therefore, Jiang et al. [231] reported the HIV-1 protease inhibitory on adenosine from *C. militaris*. Lee et al. [232] recorded the antiviral effect of *C. militaris* on DBA/2 mice infected with H1N1; the mice showed significant survival improvement following *C. militaris* treatment and marked decrease in TNF-α Kaymakci and Güler. *C. militaris*’ anti-influenza effect was confirmed by Ohta et al. [233]; they reported significant decrease in virus titers in both lung tissue and the bronchoalveolar fluid of mice, when treated with an acidic polysaccharide (APS) isolated from *C. militaris* intranasally. The anti-influenza effect of the APS is probably due to its immunomodulatory effects [232]. In addition to anti-HIV and anti-influenza activities, *C. militaris* also exhibits an anti-HCV effect [294]. They also reported that cordycepin was probably instrumental in pulling through this activity by inhibiting RNA-dependent RNA-polymerase (NS5B) in HCV [295]. *C. sinensis* and *C. militaris* can modulate immune responses as well as anti-inflammatory, antiviral, antioxidant, and antifibrotic properties. It may be suitable for the pathologies that occur in COVID-19 [296,297,298].

Another species of mushroom that has shown promising antiviral effects is *Grifola frondosa*, which has been used in herbal medicine. The major biologically active component here is β-glucan. Grifola β-glucan has shown great anticancer potential and has been approved as a therapeutic drug for cancer in China [299] and in vitro replication of HSV type 1 (HSV-1) [300]. Gu et al. (2007) confirmed that topical administration of the protein extract to the cornea of mice caused a significant decrease in virus [282]. Additionally, D-fraction from *Grifola frondosa* (GF-D), together with human IFN α-2b (IFN), was used against hepatitis b virus (HBV). Following analysis of HBV DNA and viral antigens, the results obtained showed that GF-D or IFN could control the HBV DNA in cells. Combined use of GFD and IFN synergistically inhibited HBV replication [301].

Significant increase in pro-inflammatory mediators, COX-2, pro-inflammatory cytokines TNF-α, IL-1β and IL-6 in LPS-stimulated human U937 macrophage cells is reported. The hot water extracts of P. A+ strain mushroom significantly inhibited the LPS-induced COX-2 while the other extracts lowered the levels non-significantly [302]. The study also revealed that the four hot water mushroom extracts of *Pan cyanescens*, *P. natalensis*, *P. cubensis* and *P. A+* strain significantly the two key pro-inflammatory cytokines TNF-α, IL-1β inhibited in a dose-dependent manner. Suppression of the induced IL-1β and the lowering of COX-2 following exposure to mushroom extracts indicated their potential in inflammation-related diseases. *P. natalensis* and *P. cubensis* inhibited LPS-induced IL-6 in human U937 macrophage cells. The extracts also marginally increased the concentrations of the anti-inflammatory cytokine IL-10 in the treated human macrophage cells [303,304,305]. Well-known anti-inflammatory and antioxidant compounds have been identified in *P. natalensis* [306]; other studies confirmed significant inhibition of ROS displayed by *Pan cyanescens* and *P. cubensis* in a pathological hypertrophy condition [307]. Excessive amounts of ROS stimulate the release of cytokines and subsequent activation of COX and LOXs signaling, playing a role in inflammatory reactions.

The three mushrooms *Pleurotus columbinus*, *Pleurotus sajor-caju*, and *Agaricus bisporus* contain a myriad of bioactive compounds. Aqueous extracts of these mushrooms were tested against Ad7 and HSV2 viruses. The extracts show potent antioxidant effects. *Pleurotus columbinus*, *Pleurotus sajor-caju* and *Agaricus bisporus* mushrooms offer significant medicinal potential for the prohibition and treatment of a variety of ailments [308]. Figure 2 gives an overview of the comprehensive list of viruses and the mushroom polysaccharides that have been reported to engage in antiviral activity.

### AntiCOVID-19 Activity of Mushroom Polysaccharides

The world has been suffering from the effects of the 2019 COVID-19 pandemic. As of now, limited provisions are available with respect to control, treatment and spread of COVID-19 [309,310,311]. As of now, there are few treatments available for COVID-19. The U.S. Food and Drug Administration (FDA) has approved remdesivir (Veklury) for the treatment of COVID-19. Monoclonal antibodies are laboratory-made molecules that act as substitute antibodies, by equipping the immune system to recognize and respond effectively to the virus, slowing down viral reproduction and virulence. The FDA has issued EUAs for several monoclonal antibody treatments for COVID-19 for the treatment of mild or moderate COVID-19 in adults and pediatric patients. Approved therapies using small molecules and monoclonal antibodies that have been demonstrated to be effective against COVID-19 and the proven efficacy of vaccination are also affirmed. The first pill designed to treat symptomatic COVID-19 has been approved by the UK medicines regulator as of November 2021. Molnupiravir, developed by the US drug companies Merck, Sharp and Dohme (MSD) and Ridgeback Biotherapeutics, is the first antiviral medication for COVID-19 which can be taken as a pill. Although to date there are some options, there still exists a pressing urgent need to discover novel natural antivirals that are cost-effective and exhibit enhanced anti-COVID-19 efficacy. Using an artificial intelligence (AI) programme, researchers identified components that can interfere with clathrin-mediated endocytosis and thus inhibit viral infection. These can be deployed as potential therapeutics against COVID-19 [312]. However, the problem with commercial medications is the increased risk of drug resistance development. Natural substances, such as mushrooms that have been previously discussed in the above section, clearly displayed antiviral and anti-inflammatory activity. With this as the launching pad, there is definitely scope to believe that mushrooms may hold natural remedies against COVID-19 [313]. We present the available reports in this direction.

Vilcek and Lee, in 2018 [314], elucidated the structural characterization of lentinan from *Lentinus edodes* mycelia (shiitake) and their associated anti hematopoietic necrosis virus (IHNV) potential. The novel lentinan (LNT-1) confirmed prominent antiviral activity against INHV. The antiviral mechanisms of LNT-1 were reported to be due to direct inactivation as well as inhibition of viral replication and downregulation of pro-inflammatory cytokines that are known to induce antiviral, anti-proliferative and immunomodulatory effects [315]. In case of COVID-19, the innate immune response is a critical factor for disease severity and disease outcome. COVID-19 patients exhibit high titers of inflammatory cytokines and so the effects of LNT-1 could clearly impact and lead to anti-COVID-19 acitivity [316]. Moreover, oxidative stress and inflammation are two factors that are consistently linked to the pathogenesis of COVID-19; both these factors are well within the bioactivity of mushroom based components [317].

*Inonotus obliquus* (IO) is expected to be a valuable asset against SARS-CoV-2 virus [318]. IO is well accomplished in traditional medicine, for facilitating breathing, because this mushroom has been known reduce nasopharyngeal inflammation [88,284,319]. A study demonstrating the effect of *I. obliquus* polysacharides in cats has shown to be promising, where inhibition of RNA viruses and DNA viruses was observed [320]. This mushroom inhibited viral-induced membrane fusion, and could act against the early stages of HSV viral infection. The aqueous extracts of *I. obliquus* could prevent HSV-1 entry by directly acting on viral glycoproteins, which in turn prevent membrane fusion [321]. With a host of accomplishments against various reputed viruses, IO does stand a chance against COVID, yet has a long way to go.

Spike protein and the main proteases of SARS-CoV-2 have been identified as potential therapeutic targets and their inhibition may hold the key. Nothing specific is available to treat SARS-CoV-2. Authors have established the therapeutic potential of cordycepin against COVID-19 as a conventional therapeutic strategy. Using *in silico* studies, the molecular interactions and potential binding affinity of cordycepin with SARS-CoV-2 target proteins were studied. Cordycepin is under clinical trial (NCT00709215). Attempts are being made to see if cordycepin can destabilize SARS-CoV-2 RNAs by inhibiting the polyadenylation process. This can inhibit viral replication and eventual multiplication within the host [322]. It is reported that cordycepin showed strong binding affinity with SARS-CoV-2 spike protein and main proteases that further corroborate therapeutic potential against COVID-19. Cordycepin has both pre-clinical and clinical information about antiviral activities; therefore, it is necessary that the global community tests its efficacy and safety against COVID-19. *C. sinensis* and *C. militaris* possess antiviral, immunomodulatory, and lung function protective effects, which can also be applicable for COVID-19 treatment. *C. sinensis* increased tolerance to hypoxia in the lungs by increasing Nrf2 and HIF1α and decreasing NFκB in vitro. It also increased the anti-inflammatory cytokine TGF-β [323]. *C. militaris* has an immune-enhancing effect in healthy mice and an immune-inhibitory effect in H1N1 (A/Korea/01/2009 (K/09))-infected mice. People infected with COVID-19 have high titers of inflammatory cytokines, which confirms that the lentinan polysaccharide from *L. edodes* [324] and the acidic polysaccharide (APS) of *C. militaris* should be given more attention in the fight against SARS-CoV-2. Figure 3 displays the anti-COVID-19 impacts and prospects of mushroom polysaccharides.

Clinical studies confirm the fact that β-glucans can reduce a series of symptoms of the respiratory apparatus caused by various infections, as well as the fact that they can lower systolic and diastolic artery blood pressure. It is well-known that the symptoms caused by the COVID-19 infection are severe, and studies have shown alleviation of symptoms and considerable improvement of the patient’s state following administration of β-glucans, particularly in most vulnerable cases within ICUs. This supports the fact that oral administration of β-glucans could be an efficient and inexpensive way to support the immune system of COVID-19-infected patients. However, this would require clinical confirmation *G. lucidum* to be well-known for its antitumoral, antiviral, anti-inflammatory qualities. Ganoderma is one of the most widely used in studies on antiviral qualities; it has been tested against the HIV 1 virus [287]. Table 2 consolidates the antiviral reports of mushroom polysacchraides.

Non-digestible carbohydrates with prebiotic effect, such as β-glucan polysaccharides from medicinal mushrooms, stimulate growth of gut microbes that are favorable to the host’s health and spur on the production of SCFA, which energizes anaerobic gut microbes and suppresses pathogens (e.g., *Salmonella sp.*) and improves host immunity [325,326]. In this way, mushroom polysaccharides can indirectly help patients therapeutically in the struggle against COVID-19 [327,328,329]. Additionally, with the fact that mushrooms are accomplished for their antibacterial activity, mushrooms can surely aid in the control of bacterial secondary infection (which in the second wave of COVID-19) led to high mortality. There is definitely room for input from antibacterial mushroom extracts and polysaccharides, from various angles, which needs to be incorporated positively. 

**Table 2 antioxidants-10-01899-t002:** Antiviral activity of mushroom/mushroom polysaccharides,.

Mushroom	Bioactive Component	Antiviral Activity against	IC50/CC50 Values	Reference
*Lentinus edodes*	Mannoglucan, polysaccharide–protein complex, glucan, lentinan	HSV-1; HNV	IC50: 26.69 mg·mL^−1^ to 35.12 mg·mL^−1^	[295]
*Grifola frondosa*	Proteoglycan, glucan, galatomannan, heteroglycan, and grifolan	Enterovirus 71, HSV-1	Unspecified	[283]
*Flammulina velutipes*	Glucan-protein complex, glycoprotein	Antitumor, anti-inflammatory, antiviral, immunomodulating	Unspecified	[40]
*Coriolus versicolor*	Polysaccharides PSK and PSP	Antiviral effect on HIV and cytomegalovirus in vitro and anticancer	6.25–150 μg mL^−1^	[330]
*Daedaleopsis confragosa, Datronia mollis, Ganoderma valesiacum, Irpex lacteus, Ischnoderma benzoinum, Laricifomes officinalis, Lenzites betulina, Phellinus concha-tus, Piptoporus betulinus, Trametes gibbosa, and Trametes versicolor*	Mushroom extracts	A/chicken/kurgan/05/2005 (H5N1) bird virus and the A/Aichi/2/68 (H3N2)human virus	Unspecified	[277]
*Pleurotus pulmonarius*	Mushroom water extracts	A/California/07/09 (H1N1pdm)	CC50: 1.7–8	[278]
*Pleurotus tuber-regium*	β-glucans	Herpes simplex virus type 1 (HSV-1), herpes simplex virus type 2 (HSV-2), respiratory syncytial virus (RSV) and influenza A virus (Flu A)	IC50: 3.3–6.8 μg mL^−1^	[279]
*Boletus edulis, Lentinus edodes and Pleurotus ostreatus*	Water and methanolic mushroom extracts	Herpes simplex type 1 (HSV-1)	IC50: 26.69 mg mL^−1^ to 35.12 mg·mL^−1^	[280]
*Agaricus brasiliensis*	Polysaccharide	Type 1 poliovirus	IC50: 97.2–922.9 μg mL^−1^	[281]
*Grifola frondosa*	Mushroom extracts	Enterovirus 71 and HSV-1	IC50: 4.1 μg/mL	[282,283]
*Inonotus obliquus*	Mushroom extracts	Hepatitis C virus	TCD50: 6.0 lg/mL	[284]
*Pleurotus sp. and Lentinus sp.*	Methanolic mushroom extracts	Cytomegalovirus (HCMV)	IC50: 180 μg/mL and 160 μg/mL	[285]
*Pleurotus ostreatus*	β-glucans	Influenza virus	IC50: 26.69 mg·mL^−1^ to 35.12 mg·mL^−1^	[280]
*Ganoderma lucidum*	Ganolucidic acid A, 3β-5α-Dihydroxy-6β-Methoxyergosta-7,22-Diene,ganoderic acid A–C, Ganoderic acid β, Ganodermanondiol, Ganodermanontriol and Lucidumol B	Inhibits HIV-1 protease activity	IC50: 0.17–0.23 mM	[287,288,289]
*Ganoderma colosum*	Colossolactones, ganomycin I, and ganomycin B	Anti-HIV-1 protease activity	IC50: 5–39 µg/mL	[290,291]
*Ganoderma sinnense*	Ganoderic acid GS-2, 20-hydroxylucidenic acid N, 20(21)-dehydrolucidenicacid N and ganoderiol F	Anti-HIV-1 protease activity	IC50: 22–116 μM	[290,291]
*Lignosus rhinocerus*	Crude mushroom extracts	Anti-HIV-1 protease activity	Unspecified	[292]
*Auricularia polytricha*	Ergosterol, linoleic acid and two triacylglycerols	Anti-HIV-1 protease activity	IC50: 0.80 ± 0.08 mg/mL	[292]
*Cordycep militaris*	Arabinogalactan (APS)	Anti-HIV-1 protease activity	Unspecified	[297]
*Russula paludosa*	4.5 kDa protein	Anti-HIV-1 protease activity	IC50 = 0.25 mg/mL	[293]
*Cordycep sinensis and Cordycep militaris*	cordycepin	Anti-HIV-1	Unspecified	[296,297]
*Cordycep militaris*	cordycepin	DBA/2 mice infected with H1N1	Unspecified	[232]
*Cordycep militaris*	Acidic polysaccharides (APS)	Anti-influenza	Unspecified	[324]
*Cordycep militaris*	Cordycepin	Anti-Hepatitis C Virus	Unspecified	[294]
*Grifola frondosa*	β-glucan	Inhibit in vitro replication of HSV type 1	4.1 μg/ml	[306]
*Grifola frondosa*	Protein extract	Hepatitis B virus	0.59 mg/mL and 1399 IU/ml	[306]
*Pleurotus columbinus, Pleurotus sajor-caju, and Agaricus bisporus*	Mushroom extracts	Ad7 and HSV2 viruses	Unspecified	[307]
*Grifola frondosa*	D-fraction from Grifola frondosa (GF-D)	Anti HIV	0.59 mg/mL and 1399 IU/ml	[301]
*Lentinus edodes*	Lentinan	Hematopoietic necrosis virus (IHNV)	Unspecified	[314]
*Pan cyanescens, Pan natalensis, Pan cubensis and Pan A+ strain*	Hot water mushroom extracts	Anti Cox sackievirus (COX-2)	Unspecified	[303,304,305,306]
*Inonotus obliquus*	Aqueous extract	HSV	3.82 μg/mL	[319]
*Lentinus edodes*	Lentinan	SARS-CoV-2	Unspecified	[324]
*Lentinus edodes*	Lentinan	Anti-COVID-19	Unspecified	[329]
*Inonotus obliquus*	Polysaccharides	*SARS-CoV-2 virus*	Unspecified	[324]
*Cordycep militaris*	Cordycepin	Anti SARS/Anti-COVID-19	Unspecified	[323,324]
*Cordycep sinensis*	Cordycepin	Anti SARS/Anti-COVID-19	Unspecified	[323,324]

## 5. What Is and What Is to Be

The role and explicit benefits from mushroom and mushroom polysaccharides were exploited by the ancients and this has extended to this day. As reviewed, mushrooms encompass bioactivities and unique properties and remedies that are sought after in the medical realm. The natural origin of these bioproducts in mushrooms is an added advantage. As overviewed in this paper, there is no question regarding the numerous versatile benefits that mushrooms yield. Mushrooms have come a long way and have impacted human health and wellbeing and have been there in our hearts and in our diets. Therefore, we are far still from clinical validation of these important nutrient reservoirs. Despite all the known fact files of the potentials of mushrooms, mushroom consumption is still localized to specific geographical zones. Additionally, the popular medicinal mushroom varieties are unavailable in most parts of the world. The local markets mostly are confined to button mushrooms. While the production areas are confined to the UK, Germany, Hungary, Italy, France, the consumption of varied varieties is confined to Japan, China, Korea, Taiwan, Netherlands. There is definitely a need for the sensitization of the fact that medicinal mushrooms need to be cultivated and promoted and consumed, in order to harness the full potential of this valuable natural resource. This is something that this review would like to emphasize. The mode of action/mechanism behind the bioactivity and antiviral activity of mushroom polysaccharides is far from elucidated. This review points out to the need to improve in this aspect. Understanding the fundamental modulus operandi of a bioactive material makes room for manipulation of the related aspects of components that ideally hold a position to impact their biological activity.

Mushrooms and their anti-cancer, hepato properties, antibacterial properties, antiviral properties are all known. Mushrooms applied to antiCOVID-19 research are very sparsely reported; this is another area that this review points towards for more awareness and focus. Mushroom polysaccharides are another crucial factor this review has brought up. The antioxidant activities of mushroom polysaccharides, as well as their numerous biological applications, including antibacterial and antiviral properties, have been highlighted. Mushroom polysaccharides have been scarcely highlighted. With the known facts regarding their versatility, this review calls for attention on the area of mushroom polysaccharide research. Additionally, we project a concern, which needs to be looked into—how much of these polysaccharides we are losing during our cleaning process, prior to cooking. This is an aspect this review is critical about. With many water-soluble polysaccharides and extracellular polysaccharides around, the compromises that the processing food industry and domestic cooking processes are making, leading to the loss of this valuable component, are worth probing. Biologically active polysaccharides are widespread among higher basidiomycetous mushrooms, and most of them have unique structures in different species. These polysaccharides have different compositions, most belonging to the group of -d-glucans; these have -(1–3) linkages in the main chain of glucan and additional -(1–6) branch points. High molecular weight glucans apparently seem to be more effective than those of low molecular weight. Moreover, different strains can produce polysaccharides with different properties. For example, the proteoglycan Krestin was developed in Japan from the strain Trametes (Coriolus) versicolor CM-101, whereas a polysaccharide–peptide (PSP) in China was developed in submerged culture from the Cov-1 strain of the same species [330].

With the known importance of mushroom polysaccharides, amplifying the genes that govern biosynthesis of mushroom polysaccharides using molecular engineering can be a very resourceful direction [331]. Chai et al. [332] demonstrated overproduction of β-glucans in *Pleurotus ostreatus* mushrooms by promoter engineering. The promoter for the 1,3-β-glucan synthase gene was replaced by the promoter of glyceraldehyde-3-phosphate dehydrogenase gene of *Aspergillus nidulans*, leading to enhanced β-glucan yield compared to the wild type. Ji et al. [333] improved polysaccharide production by bioengineering the biosynthetic pathway in *Ganoderma lucidum.* Overexpression of the homologous UDP glucose phosphoglucomutasegene leads to near doubling of the intracellular and extracellular polysaccharides contrasted to wild type. Meng et al. [208] used the Viteoscilla hemoglobin gene to increase extracellular and intracellular polysaccharides in *G. lucidum* [334]; amplification of these genes might induce the formation of high-polysaccharide mushrooms [334]. Khan et al. [335] reported that irradiation with 50 k Gy doses induced bond cleavage, enhanced antioxidant activity and increased functional properties. Except for these pioneering reports on the potential of gene manipulation to enhance the production of polysaccharides, successful implementation of antioxidants’ mushroom polysaccharide still remains insufficiently explored. There is so much that is known and has been established, yet we are so far from practical implementation. This is something that we highlight as a future perspective in this area. Mushroom culturing needs expansion and awareness regarding the right choice of mushrooms that need to be propagated. Most of the culturing techniques are the age-old methods; the rightful improvisations with culture techniques that may promote and accelerate polysaccharide production, are grey areas, which when worked upon can extend the full exploitation of this resource.

## 6. Conclusions

The bioactivity of mushrooms with specificity to mushroom polysaccharides has been reviewed. The antioxidant properties of mushroom polysaccharides and their antioxidant mechanisms have been discussed. The need to extrapolate the existing beneficial attributes into the current pandemic scenario has been emphasized. Mushrooms as a natural remedy for COVID-19 are still inadequately addressed. This review discusses the lacunae in this area of research and highlights aspects that need attention. When the world is looking for answers to the COVID-19 pandemic, we might have some valuable help just in the area of mushrooms.

## Figures and Tables

**Figure 1 antioxidants-10-01899-f001:**
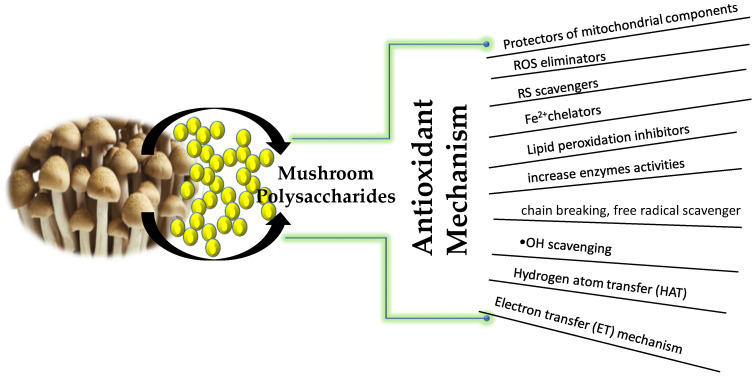
Antioxidant mode of action of mushroom polysaccharides.

**Figure 2 antioxidants-10-01899-f002:**
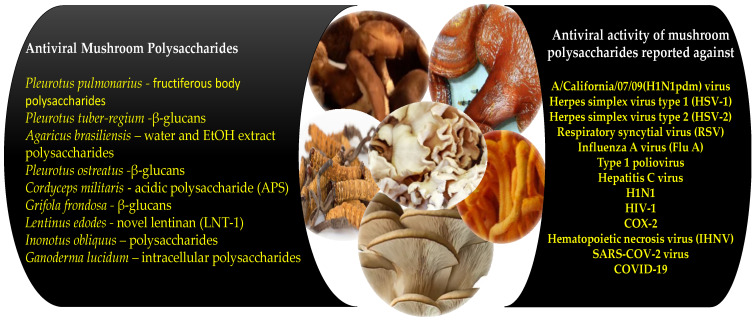
Overview of antiviral mushroom polysaccharides and impacted viruses.

**Figure 3 antioxidants-10-01899-f003:**
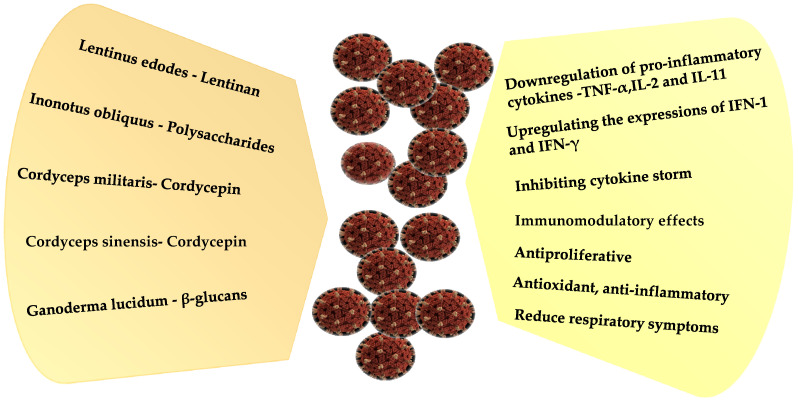
AntiCOVID-19 impacts of mushroom polysaccharides.

**Table 1 antioxidants-10-01899-t001:** Bioactivity of Mushroom Polysaccharides.

Source Mushroom	Bioactive Component	Antioxidant Activity	References
*Agaricus brasiliensis*	Crude Se polysaccharide	Scavenging of DPPH and hydroxyl radicals	[272]
*Phellinus xiaobaumii*	Homogenous water soluble polysaccharide	Hydroxyl, superoxide and DPPH radical scavenging	[274]
*Pleurotus abalonus*	Polysaccharide–peptide complex LB-1b	Antioxidant activity in erythrocyte haemolysis	[275]
*Cordyceps taii*	Polysaccharides	DPPH, hydroxyl, and superoxide anion radical scavenging	[268]
*Agaricus bisporus*	Polysaccharides	Free radical scavangers enhancement of antioxidant enzymes in sera, liver, and heart of mice	[276]
*Ganoderma lucidum*	Heteroglycan, mannoglucan, glycopeptide	Antioxidant	[34]
*Pleurotus ostreatus*	Glycoprotein	Antitumor, hyperglycemia, antioxidant	[34]
*Cordiceps militaris*	Polysaccharides	Antioxidant activity suppression of in vivo growth of melanoma in mouse models	[251,252]
*Inonotus oblique*	Crude polysaccharide	Used as an antioxidant in Russian folk medicine	[253]
*Hericium erinaceus*	Unique polysaccharide EP-1	Strong in vitro antioxidant activity in mice	[254]
*Macrolepiota dolichaula*	Fucogalactan	Antioxidant and immunostimulating properties in vitro	[189]
*Russula albonigra*	β-glucan	Antioxidant and immunostimulating properties in vitro	[255]
*Tricholoma mongolicum*	Folysaccharides	In vitro antioxidant activities	[202]
*Ganoderma*	β-d-glucans	In vitro antioxidant activity	[259]
*Entoloma lividoalbum*	Water soluble β-glucan	High antioxidant activity	[260]
*Tylopilus ballouii*	Fucogalactomannan	Inhibiting superoxide and hydroxyl radicals	[261]
*Ganoderma lucidum*	α- and β-glucans	High antioxidant activity	[262]
*Fistulina hepatica, Pleorotus squarrosulus* *, Polyporus grammocephalus* *, Phellinus linteus* *, Austreus hygrometricus, and Macrocybe gigantea*	Polysaccharides	Significant antioxidant potential	[263]
*Ganoderma tsugae*	Polysaccharides	Best scavenging activity	[268]
*Ganoderma lucidum* *, Hypsizygus marmoreus, Pleurotus ostreatus, Pleurotus nebrodensis, Lentinus edodes, Pleurotus eryngii, Flammulina velutipes, and Hericium erinaceus*	Polysaccharide activity compared	Significant antioxidant potential	[249]

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
