# Peer review of "Antioxidant Activity of Mushroom Extracts/Polysaccharides—Their Antiviral Properties and Plausible AntiCOVID-19 Properties"

_antioxidants, 2021, doi:10.3390/antiox10121899_

Round 1

Reviewer 1 Report

Dear Editor,

The manuscript submitted by Sechul Chun et al. and entitled: “Antioxidant activity of Mushroom polysaccharides – Antiviral  properties and plausible AntiCOVID?” is a very interesting work given a good overview of antioxidant polysaccharides from mushroom. This manuscript can be accepted after major revisions. Please to see my comments below.

Comments:

  • Authors should reformulate the title of the revised manuscript for a better understanding of readers. In fact, I think authors should not mention the Anticovid in the title. Because works are still in progress regarding the real impact of polysaccharides for in vivo “Covid”
  • In the whole manuscript, lot of descriptions on anti-covid have been stated without real study of action’s mechanism occurred in the interaction between mushroom polysaccharides and virus. Authors should reformulate all these description using good description.

General comment:

  • In the revised manuscript, the authors need to pay more attention to grammatical construction of sentences and spelling of sentences.
  • The quality of some figures and tables should be improved in the revised manuscript.
  • In the conclusion part, authors must give more details about the aspects of novelty and the food/therapeutic applications. These parts have to be more underlined in relation with other known bioactive polysaccharides from plants, bacteria, algae et animals.

Author Response

The manuscript submitted by Sechul Chun et al. and entitled: “Antioxidant activity of Mushroom polysaccharides – Antiviral  properties and plausible AntiCOVID?” is a very interesting work given a good overview of antioxidant polysaccharides from mushroom. This manuscript can be accepted after major revisions. Please to see my comments below.

Ans. First of all we would like to thank the Editor and the reviewers for their time and input into our manuscript. We are extremely grateful for the revision opportunity. We also acknowledge the encouraging comments and remarks on the paper. There is no doubt each of your suggestions implemented into the manuscript will increase the quality of our paper. We appreciate your comments. We have now revised the manuscript based on your comments. We have responded to your comments point by point and highlighted the changes made to the manuscript using track changes. Thank you again.

Comments:

  • Authors should reformulate the title of the revised manuscript for a better understanding of readers. In fact, I think authors should not mention the Anticovid in the title. Because works are still in progress regarding the real impact of polysaccharides for in vivo “Covid”

Ans. Yes you are right in that nothing much has been done in the direction of COVID, that is why we are wording it ‘Antiviral properties and plausible AntiCOVID?’ with a question mark. When tons of work has been done with respect to mushroom polysaccharides and viruses, why not COVID is the point highlighted here. Thank you for your understanding.

  • In the whole manuscript, lot of descriptions on anti-covid have been stated without real study of action’s mechanism occurred in the interaction between mushroom polysaccharides and virus. Authors should reformulate all these description using good description.

Ans. Very thoughtful query indeed, yes no report is complete without touching on the mechanism/modulus operandi, the issue here is that not much is established in terms of working out the mechanism of action of mushroom polysaccharides and antiviral activity. We have now addressed the need to expand on this aspect as future direction under Section 5.  Thank you.

General comment:

  • In the revised manuscript, the authors need to pay more attention to grammatical construction of sentences and spelling of sentences.

Ans. We have now. Thank you. Sorry for the inconvenience.

  • The quality of some figures and tables should be improved in the revised manuscript.

Ans. We have now improved on the quality.

  • In the conclusion part, authors must give more details about the aspects of novelty and the food/therapeutic applications. These parts have to be more underlined in relation with other known bioactive polysaccharides from plants, bacteria, algae et animals.

Ans. As you can see the manuscript as such is quiet long, we have restricted to the focused topic of mushrooms Polysaccharides and viruses. This way we could achieve more depth. Thank you. 

Reviewer 2 Report

Authors did a huge work for collecting and organising the different antioxidant and potential antiviral activities of mushrooms. Before acceptance I would strongly suggest to clarify and ann details to most of the bioactive compounds. I would be great not only to see "polisaccharydes", but exact names of the active agents. 

In addition there are some review papers that are very similar. Although some of these are cited in the manuscript, I suggest to check similarities, check whether these data are already included or the manuscript might need some addition, and in some cases I suggest to add actual bioactivities (IC50 etc) to the tables at least in the form of intervals e.g. 1-100 uM.

https://www.mdpi.com/2071-1050/13/14/7948/htm https://www.ncbi.nlm.nih.gov/pmc/articles/PMC3382807/ https://www.ncbi.nlm.nih.gov/pmc/articles/PMC7551890/ https://www.ncbi.nlm.nih.gov/pmc/articles/PMC8399653/ 

Author Response

Authors did a huge work for collecting and organising the different antioxidant and potential antiviral activities of mushrooms. Before acceptance I would strongly suggest to clarify and ann details to most of the bioactive compounds. I would be great not only to see "polisaccharydes", but exact names of the active agents. 

Ans. Thank you very much for the encouraging appreciation. We have now provided names of the polysaccharides, where ever possible and whichever available. Some of the papers only report crude polysaccharides unspecified. Those we retain as such. Thank you.

In addition there are some review papers that are very similar. Although some of these are cited in the manuscript, I suggest to check similarities, check whether these data are already included or the manuscript might need some addition, and in some cases I suggest to add actual bioactivities (IC50 etc) to the tables at least in the form of intervals e.g. 1-100 uM.

Ans. Yes we understand your concern. We have now cited the reviews in this area and also added the IC50 values, wherever possible in the revision. Thank you.

https://www.mdpi.com/2071-1050/13/14/7948/htm https://www.ncbi.nlm.nih.gov/pmc/articles/PMC3382807/ https://www.ncbi.nlm.nih.gov/pmc/articles/PMC7551890/ https://www.ncbi.nlm.nih.gov/pmc/articles/PMC8399653/ 

Ans. Added these citations. Thank you.

Reviewer 3 Report

The manuscript of Chun and colleagues is addressed to review the antioxidant activity of mushroom polysaccharides and their potential use to develop therapies against COVID-19.

However, the involvement of polysaccharides in the antiviral strategies development is poorly reported in the manuscript and few data support their use against COVID-19.

Some section of the manuscript are well written, but some sections require a relevant revision (such as sentences from the line 263 to line 269, or the line 309, or lines 428-431 and so on). The sentences starting from lines 454 and 530 are wrong and other sentences need to be modified.

Change the word “antibiotic” in the line 174, it is not appropriate with viruses. An option is “antimicrobial”.

Nomenclature require a revision. For example instead of H1N1 to indicate an influenza virus is important report the full name of the strain. Some taxonomic names are not reported in italics.

References are not correct, there are probably mistakes and some references are not correlated to the concepts reported in a sentence (for example [368] line 478, the reference is about HBV and not HSV, or [390] on flu virus, and other references).

In addition, authors want to focalize their review paper on polysaccharides. However, in the paragraph 4 “Antiviral activity of Mushrooms/mushroom polysaccharides” are reported data on antiviral activities of other biological molecules, as proteins (see line478), terpenes (lines 450-453) and other molecules (see lines 457-458) or unrelated to virus infections (see lines 494-500). It is a mix of compounds/info with poor specific info on polysaccharides.

The description of the antiviral activity should be more detailed specifying if studies are performed in vivo or in vitro.

Furthermore, sentences about COVD-19 pandemic require to be updated (lines 508-516). Today there are approved therapies with small molecules and monoclonal antibodies and the efficacy of vaccination is clearly evident!

In the tables, the name of mushrooms is reported as full name or in the short format. The same format should be used.

Author Response

The manuscript of Chun and colleagues is addressed to review the antioxidant activity of mushroom polysaccharides and their potential use to develop therapies against COVID-19. However, the involvement of polysaccharides in the antiviral strategies development is poorly reported in the manuscript and few data support their use against COVID-19.

Ans. First of all we would like to thank the Editor and the reviewers for their time and input into our manuscript. We are extremely grateful for the revision opportunity. We also acknowledge the encouraging comments and remarks on the paper. There is no doubt each of your suggestions implemented into the manuscript will increase the quality of our paper. We appreciate your comments. We have now revised the manuscript based on your comments. We have responded to your comments point by point and highlighted the changes made to the manuscript using track changes.

Actually, this review aims at pointing out to the fact that mushroom polysaccharides have been poorly reported and tested with respect to COVID. The facts that mushroom polysaccharides are reputed for their bioactivity and antiviral properties and yet this technology transfer for COVID that has not been applied, is highlighted and emphasized in the future perspective.

Some section of the manuscript are well written, but some sections require a relevant revision (such as sentences from the line 263 to line 269, or the line 309, or lines 428-431 and so on). The sentences starting from lines 454 and 530 are wrong and other sentences need to be modified.

Ans. We have revised the sections mentioned. Thank you.

Change the word “antibiotic” in the line 174, it is not appropriate with viruses. An option is “antimicrobial”.

Ans. Changed. Sorry about that. Thank you for rightly pointing out.

Nomenclature require a revision. For example instead of H1N1 to indicate an influenza virus is important report the full name of the strain. Some taxonomic names are not reported in italics.

Ans. We apologize for these, we have revised these. Thank you.

References are not correct, there are probably mistakes and some references are not correlated to the concepts reported in a sentence (for example [368] line 478, the reference is about HBV and not HSV, or [390] on flu virus, and other references).

Ans. Corrected. We also checked all the references.  Sorry for that.

In addition, authors want to focalize their review paper on polysaccharides. However, in the paragraph 4 “Antiviral activity of Mushrooms/mushroom polysaccharides” are reported data on antiviral activities of other biological molecules, as proteins (see line478), terpenes (lines 450-453) and other molecules (see lines 457-458) or unrelated to virus infections (see lines 494-500). It is a mix of compounds/info with poor specific info on polysaccharides.

Ans. We have now revised this section and removed unnecessary controversial stuff from it. Thank you.

The description of the antiviral activity should be more detailed specifying if studies are performed in vivo or in vitro.

Ans. Added this information wherever possible. Thank you.

Furthermore, sentences about COVD-19 pandemic require to be updated (lines 508-516). Today there are approved therapies with small molecules and monoclonal antibodies and the efficacy of vaccination is clearly evident!

Ans. Updated. Thank you.

In the tables, the name of mushrooms is reported as full name or in the short format. The same format should be used.

Ans. Very sorry about the incoherent format style. We have now revised it. Thank you.

Round 2

Reviewer 1 Report

Corrections have been done according to reviewers's remarks.

Consequently, this paper could be accepted in current form.

Regards

Author Response

Corrections have been done according to reviewers's remarks.Consequently, this paper could be accepted in current form.Regards

Ans. We would like to thank our dear reviewer for the kind acknowledge of our efforts towards revising the manuscript. Thank you that you have accepted our labour.

Reviewer 3 Report

The new version of the manuscript has been improved only for some details.  However, it is very poor about the key concept on polysaccharides as antivirals and many ather molecules were reported as antivirals. The title of the review paper is not adeguate with the content of the article. If the knowledge of polysaccharides is not enough a review paper is not appropriate but a different type of manuscript should be selected (editorial, opinion paper, hypothesis paper, ...).

Some updated were introduced as request, however the manuscript was not properly adapted, such as the paragraph 4.1.  It starts as "The world has been groaning under the sway of the 2019 pandemic. As on date, no specific treatment for COVID-19 is authenticated;...." But this is in contrast with the following new sentence "Inspite of the approved therapies using small molecules and monoclonal antibodies that have been demonstrated against COVID and the proven efficacy of vaccination,..."

Futhermore, what do you mean with the word "Regime" in the line 21?

There are some mistakes in the format of the name of taxonomic words such as lines 21-22, and others).

Several acronyms have never been defined in the text (e.g. SOD; CAD; ect. see line 131). 

Author Response

The new version of the manuscript has been improved only for some details.  However, it is very poor about the key concept on polysaccharides as antivirals and many ather molecules were reported as antivirals. The title of the review paper is not adeguate with the content of the article. If the knowledge of polysaccharides is not enough a review paper is not appropriate but a different type of manuscript should be selected (editorial, opinion paper, hypothesis paper, ...).

Ans. Thank you for all your valuable suggestions and second round of comments on our manuscript. We do understand that it’s a privilege that you have meticulously working with us towards upgrading the manuscript. Thank you again.

About the concern that there is not much content on mushroom polysaccharides, as you can see from section 3 there is definitely some decent work done. But, we have considered your opinion and have reworded the title to broaden the subject area. Thank you.

 Revised Title : Antioxidant activity of Mushroom extracts/polysaccharides – their antiviral properties and plausible antiCOVID?

Some updated were introduced as request, however the manuscript was not properly adapted, such as the paragraph 4.1.  It starts as "The world has been groaning under the sway of the 2019 pandemic. As on date, no specific treatment for COVID-19 is authenticated;...." But this is in contrast with the following new sentence "Inspite of the approved therapies using small molecules and monoclonal antibodies that have been demonstrated against COVID and the proven efficacy of vaccination,..."

Ans. True these sentences don’t flow together, we have now modified this.

Thank you.

Futhermore, what do you mean with the word "Regime" in the line 21?

Ans. Since you are not comfortable with it we have removed it and reworded that. Thank you.

There are some mistakes in the format of the name of taxonomic words such as lines 21-22, and others).

Ans. We have corrected these. Thank you.

Several acronyms have never been defined in the text (e.g. SOD; CAD; ect. see line 131). 

Ans. Sorry about that, you are right, we have now defined them in the text, thank you for your patience.

Round 3

Reviewer 3 Report

Authors modified the manuscript as requested and the quality is improved.

No further comments are arised.